# Blood and brain gene expression signatures of chronic intermittent ethanol consumption in mice

Laura B. Ferguson[1,2,3]*, Amanda J. Roberts[4], R. Dayne Mayfield[1,3], Robert O. Messing[1,2,3]*

1 Waggoner Center for Alcohol and Addiction Research, University of Texas at Austin, Austin, Texas, United States of America, 2 Department of Neurology, Dell Medical School, University of Texas at Austin, Austin, Texas, United States of America, 3 Department of Neuroscience, University of Texas at Austin, Austin, Texas, United States of America, 4 Animal Models Core Facility, The Scripps Research Institute, San Diego, California, United States of America

* lferguson@utexas.edu (LBF); romessing@austin.utexas.edu (ROM)

**Data Availability Statement:** Raw data (fastq files) and processed data (raw counts and log-CPM

## Abstract

Alcohol Use Disorder (AUD) is a chronic, relapsing syndrome diagnosed by a heterogeneous set of behavioral signs and symptoms. There are no laboratory tests that provide direct objective evidence for diagnosis. Microarray and RNA-Seq technologies enable genome-wide transcriptome profiling at low costs and provide an opportunity to identify biomarkers to facilitate diagnosis, prognosis, and treatment of patients. However, access to brain tissue in living patients is not possible. Blood contains cellular and extracellular RNAs that provide disease-relevant information for some brain diseases. We hypothesized that blood gene expression profiles can be used to diagnose AUD. We profiled brain (prefrontal cortex, amygdala, and hypothalamus) and blood gene expression levels in C57BL/6J mice using RNA-seq one week after chronic intermittent ethanol (CIE) exposure, a mouse model of alcohol dependence. We found a high degree of preservation (rho range: [0.50, 0.67]) between blood and brain transcript levels. There was small overlap between blood and brain DEGs, and considerable overlap of gene networks perturbed after CIE related to cell-cell signaling (e.g., GABA and glutamate receptor signaling), immune responses (e.g., antigen presentation), and protein processing / mitochondrial functioning (e.g., ubiquitination, oxidative phosphorylation). Blood gene expression data were used to train classifiers (logistic regression, random forest, and partial least squares discriminant analysis), which were highly accurate at predicting alcohol dependence status (maximum AUC: 90.1%). These results suggest that gene expression profiles from peripheral blood samples contain a biological signature of alcohol dependence that can discriminate between CIE and Air subjects.

## Author summary

Recent evidence in mice suggests that brain gene expression profiles can predict disease status as well as predict drugs effective for treating alcoholism. However, it is not possible

voom-transformed normalized data) are publicly available on Gene Expression Omnibus GSE176122. R code used to analyze data can be found at https://github.com/zeavin-ferguson/blood_brain. Data can be explored at https://lauraferguson.shinyapps.io/blood_brain/.

**Funding:** This work was supported by National Institute on Alcohol Abuse and Alcoholism grants F32AA028148 (LBF), R01AA012404 (RDM), U01AA020926 (RDM), U01AA013520 (ROM), and R01AA026075 (ROM). The funders had no role in study design, data collection and analysis, decision to publish, or preparation of the manuscript.

**Competing interests:** The authors have declared that no competing interests exist.

to obtain brain specimens from human patients which limits the usefulness of this approach. This study investigated the extent to which blood can act as a surrogate for brain tissue and predict CIE-induced alcohol dependence status. This information lays critical groundwork for developing molecular-based diagnosis and treatment options for alcoholic patients and provides insights into the biological mechanisms that might contribute to the transition from recreational alcohol use to excessive drinking.

## Introduction

Alcohol Use Disorder (AUD) is a highly prevalent and costly syndrome with few effective treatments [1–3]. AUD like other psychiatric disorders is diagnosed by evaluating a patient's symptoms and behaviors over time as described in the Diagnostic and Statistical Manual of Mental Disorders (DSM5) [4]. Patients meeting two or more criteria within the last year are considered to have AUD with different degrees of severity. Adding molecular-based criteria would provide useful objective data to refine diagnosis and possibly afford earlier detection of problematic drinking before detrimental medical, legal, or social consequences of AUD appear. There are currently three FDA-approved treatments for AUD: disulfiram, acamprosate, and naltrexone, none of which are effective for all patients. Moreover, there are no reliable prognostic indicators that predict responses to therapeutic intervention.

Improvements in molecular technologies, computing power, and bioinformatics have revolutionized many fields of science and are beginning to impact medicine by harnessing vast amounts of data to inform diagnosis, prognosis, and treatment. Many genome-wide gene expression datasets are available from different brain regions, multiple species, and alcohol-related phenotypes. Studies examining these datasets have revealed that alcohol use (or the genetic risk for excessive alcohol use) alters brain gene expression, and these alterations can distinguish alcohol dependent subjects from healthy individuals, as well as predict therapeutic compounds. To discriminate between AUD and control subjects, application of partial least squares discriminative analysis (PLSDA) to gene expression patterns from postmortem prefrontal cortex tissue [5] has revealed a consistent re-programming of gene expression by years of having AUD that reliably discriminates AUD from non-AUD individuals. Our group recently identified a gene expression signature of risk for binge drinking from the brains of HDID-1 mice selected for high levels of binge alcohol drinking, and then used the Library of Integrated Network-Based Cellular Signatures (L1000) database from the Broad Institute to identify drugs with opposing patterns of gene expression, hypothesizing that drugs that produce anti-correlated patterns of gene expression might reduce alcohol drinking. The top-ranking drug candidates, terreic-acid and pergolide, both reduced ethanol consumption and blood alcohol levels in HDID-1 mice [6]. While these examples are promising, they have relied on *brain* gene expression data, and it is not possible to get brain tissue from AUD patients. Thus, applying advanced computational approaches to diagnose AUD and personalize AUD treatment will require noninvasive access to biological samples, such as blood. To this end, it is important to understand the extent to which blood can be used as a surrogate tissue for brain.

Although AUD is primarily considered a brain disease, alcohol use affects multiple other tissues and systems including gut, liver, lung, muscle, bone, heart, blood vessels, pancreas, and the immune system [7]. Whole blood is readily available and is routinely obtained in the clinic. We hypothesized that blood could be a useful surrogate for brain tissue, because it contacts every organ in the body, including the brain. Blood expresses ~80% of the genes that are expressed in brain, most of which are responsive to physiological or environmental

adaptations [8]. These genes include most of those that have been linked to AUD through genome-wide association studies or to alcohol-related behaviors in preclinical rodent studies [9, 10]. Blood and brain genome-wide gene expression profiles have been compared for several brain diseases including, schizophrenia [11], bipolar disorder [12], depression, [13, 14] Huntington's disease [15] and other neurodegenerative disorders [16], autism [17], and PTSD [18, 19]. These profiles have not been compared for AUD. To fill this knowledge gap, we compared gene expression levels in whole blood to those in the amygdala, prefrontal cortex, and hypothalamus from CIE mice undergoing withdrawal. We chose these brain areas because of their importance in reward signaling and alcohol dependence [20, 21] and because gene expression responses in these brain areas have been shown to parallel those in blood under other conditions [11, 15, 19, 22–24]. We used the chronic intermittent ethanol (CIE) procedure [25] to induce alcohol dependence in male and female C57BL/6J mice. We used this strain of mice because they show significant increases in alcohol drinking after being made alcohol dependent through CIE exposure [26–28]. CIE is thought to model the alcoholic individual's experience of episodic patterns of excessive ethanol consumption with repeated withdrawals [27]. It is postulated that alcohol is initially ingested for its rewarding effects, but repeated use leads to a negative emotional state experienced in alcohol's absence (the "dark side" of addiction) [29]; this state promotes excessive alcohol drinking through negative reinforcement, and treatments targeted against this state are postulated to help prevent relapse [30]. After CIE, rodents exhibit increased stress-responsiveness, depression-like behavior, and anxiety-like behavior [31–35], and they show a persistent increase in voluntary alcohol consumption that results in high blood alcohol levels (BALs) [26, 27, 36, 37]. We assessed gene expression profiles after 1 week of alcohol withdrawal because this is when increased voluntary alcohol intake is highest [27]. Additionally, the assessment of longer-term gene expression effects of ethanol avoided the confounds of acute ethanol intoxication and the volatile gene expression changes observed in early withdrawal [38, 39]. The within-subjects design we used to compare blood and brain gene expression enabled the detection of correlated responses across both tissues during withdrawal.

## Results

### Chronic intermittent ethanol (CIE) effects on voluntary ethanol drinking

To induce alcohol dependence, C57BL/6J mice underwent chronic intermittent ethanol (CIE) exposure interspersed with voluntary drinking sessions as described in the Materials and Methods section. For male mice, there was a main effect of treatment ($F_{(1, 16)} = 6.463$, $p = 0.0217$) and time ($F_{(4, 64)} = 12.73$, $p < 0.0001$), and a time x treatment interaction ($F_{(4, 64)} = 4.830$, $p = 0.0018$). Planned Dunnett's comparisons revealed that CIE significantly increased voluntary intake of 15% ethanol over baseline after CIE cycles 2, 3, and 4 (Fig 1). Ethanol consumption in mice exposed to air did not differ from baseline drinking levels throughout the study, except for a slight decrease in ethanol intake after the first air exposure (Fig 1).

For female mice, a two-way ANOVA showed a main effect of time ($F_{(4, 68)} = 5.882$, $p = 0.0004$), but not of treatment ($F_{(1, 17)} = 2.745$, $p = 0.1159$) and no time x treatment interaction ($F_{(4, 68)} = 1.436$, $p = 0.2317$). A one-way ANOVA revealed a main effect of time for the mice receiving ethanol vapor ($F_{(4, 45)} = 3.677$, $P = 0.0113$) but not air ($F_{(4, 40)} = 1.466$, $P = 0.2306$). Planned Dunnett's comparisons revealed that CIE significantly increased voluntary intake of 15% ethanol over baseline after CIE cycles 2, 3, and 4 for the ethanol treatment group (Fig 1). Ethanol consumption in mice exposed to air did not differ from baseline drinking levels throughout the study.

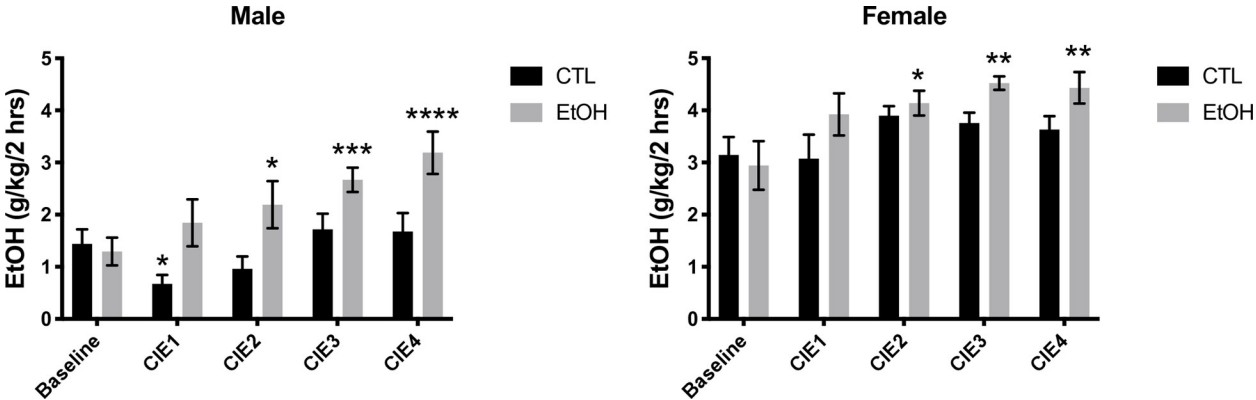

**Fig 1. Effects of CIE on voluntary drinking in C57Bl/6J mice.** Chronic intermittent ethanol (CIE) exposure significantly increased voluntary ethanol (15%) intake in male (but not female) mice as revealed by a two-way ANOVA. For female mice a one-way ANOVA revealed an effect of time on voluntary ethanol consumption in only the ethanol vapor group (EtOH) and not the control group (CTL). Results of Dunnett's planned comparisons are indicated above the SEM bars. $^*p < 0.05$, $^{**}p < 0.01$, $^{***}p < 0.001$, and $^{***}p < 0.0001$ vs the baseline group. Values represent mean±SEM (n = 8-10/sex/group).

## Comparison of blood and brain gene expression

A major goal of this study was to determine the correspondence between peripheral blood and brain transcriptomes. We addressed this in several complimentary ways. First, to determine if the transcriptional response to CIE in blood are reflective of those in brain, we assessed the overlap of the genes and gene networks affected by CIE exposure across tissues. Next, to determine whether blood transcript levels can predict those in brain irrespective of treatment, we included all subjects and calculated the correlation coefficient between blood and brain gene expression levels.

## CIE-responsive genes conserved between blood and brain

We found that CIE exposure dysregulated the expression levels of hundreds of genes in brain and blood (Fig 2A and S1 Table). The number of DEGs in brain was greater in males than females, while females showed a greater number of DEGs in blood relative to males (Fig 2A). To gain insight into the cellular specificity of gene expression perturbations, we determined whether any cell type-specific genes were over-represented in each of the gene sets. In females, microglial genes were up-regulated in all brain areas, endothelial genes up-regulated in amygdala and hypothalamus, and neuronal, T cell, and macrophage genes up-regulated in hypothalamus (Fig 2A and S2 Table). Neuronal and oligodendrocyte genes were down-regulated in female amygdala and hypothalamus (Fig 2A and S2 Table). In males, there were no cell type specific genes enriched in any of the up-regulated gene sets (Fig 2A and S2 Table). Astrocytic genes were down-regulated in male amygdala and hypothalamus, oligodendrocyte genes were down-regulated in amygdala, and microglial and endothelial cell genes were down-regulated in PFC (Fig 2A and S2 Table). In whole blood, T cell and B cell genes were down-regulated in females, while macrophage, neutrophil, and monocyte genes were down-regulated in males (Fig 2A and S2 Table).

We compared the DEGs in whole blood to those in each brain region. Depending on the brain region and sex, there were 45 to 88 overlapping DEGs between blood and brain. This overlap was statistically significant between whole blood and hypothalamus and PFC in male mice, and borderline significant between whole blood and hypothalamus (p = 0.064) and amygdala (p = 0.046) in female mice (Fig 2B).

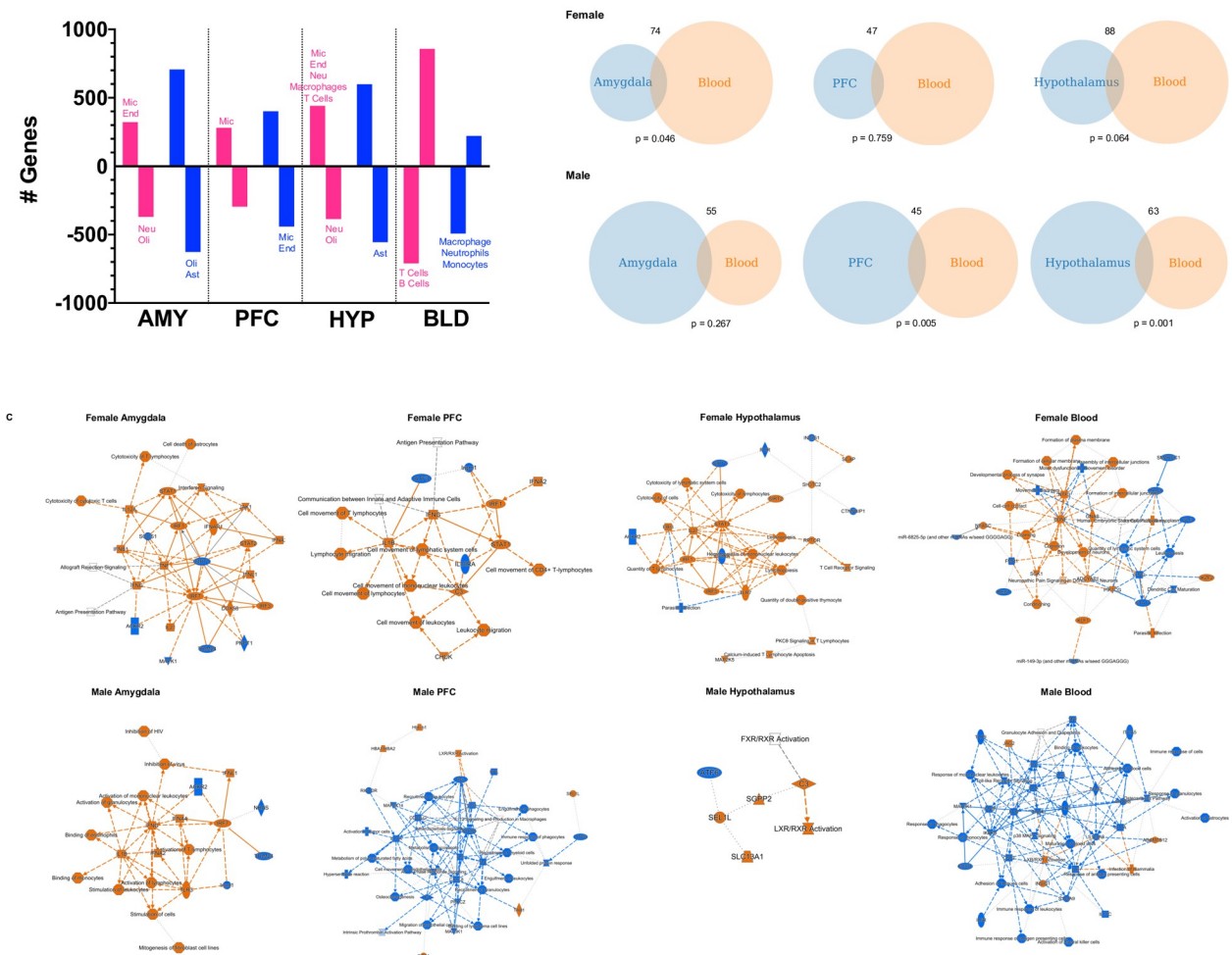

**Fig 2. Differentially expressed Genes between CIE and Air C57BL/6J mice.** (A) Chronic intermittent ethanol (CIE) exposure affected gene expression levels one week after the last air or ethanol vapor exposure in brain and whole blood in male (blue) and female (pink) mice. The bar plot shows the number of up-regulated and down-regulated genes in each tissue for each sex. Males show more differentially expressed genes in brain, while females show more differentially expressed genes in blood. If there were any significantly enriched cell type specific genes in the datasets (hypergeometric test; Bonferroni-corrected p < 0.05), these are indicated adjacent to the respective dataset. No label indicates that there were no cell type specific genes enriched in the dataset. Cell type specific datasets were derived from the literature [122, 123]. (B) The overlap between the differentially expressed genes (DEGs) in blood and brain. DEGs between CIE versus Air mice were identified using the *limma* moderated t statistic in R (nominal p < 0.05). Overlap significance was assessed with the hypergeometric test (the p-value is shown beneath the Venn diagram). (C) Graphical Summaries of the IPA Core Analysis provide an overview of the major biological themes in the DEGs and illustrate how these concepts relate to one another. This feature selects and connects a subset of the most significant entities predicted in the analysis, including canonical pathways, upstream regulators, diseases and biological functions. The algorithm constructs the summary using machine learning techniques to prioritize and connect entities and infers relationships to connect entities not yet connected by findings in the QIAGEN Knowledge Graph. These inferred relationships help to visualize related biological activities. To be included in the graph each entity must be significant (Fisher Exact test p < 0.05). Diseases, functions, and upstream regulators must also include a z score magnitude of 2 or greater (the z score is the predicted direction of effect on the pathway). Orange indicates a predicted activation and blue a predicted inhibition of the entity. The color intensity represents the strength of the z score prediction. AMY = amygdala, PFC = prefrontal cortex, HYP = hypothalamus, BLD = blood, Mic = microglia, End = brain endothelial cells, Neu = neurons, Oli = oligodendrocytes, Ast = astrocytes.

To gain functional insight into these gene perturbations, we performed a core analysis on the DEGs in each tissue using Ingenuity Pathway Analysis software. The full results are reported in S3 Table. Graphical Summaries of the IPA Core Analysis provided an overview of the major biological themes in the DEGs (Fig 2C). The Graphical Summaries depict a subset of the most significant entities predicted in the analysis (including canonical pathways, upstream

regulators, diseases, and biological functions) and their relationships between one another. Immune-related entities were prominent in all the gene sets and were generally predicted to be activated in brain and inhibited in blood (except in male PFC where immune-related entities also were predicted to be inhibited). For example, *Ifng* and *Stat1* were predicted to be activated in all female brain regions but inhibited in blood, and *Il33*, *Il17a*, and *Map3k1* were predicted to be inhibited in male blood and PFC (Fig 2C). Irf7 was a "hub" entity predicted to be activated in the amygdala of both sexes (Fig 2C). There were also several categories related to leukocyte extravasation and activation (e.g., recruitment of leukocytes, leukocyte migration, cell movement of leukocytes) which followed a similar pattern as the immune-related entities (activated in brain and inhibited in blood and male PFC) (Fig 2C). In addition to the predicted inhibition of immune-related entities in female blood, there were a number of entities that have been traditionally been associated with brain that were predicted to be activated in female blood, e.g., learning, cognition, *Bdnf* as "hub" entity (Fig 2C). Nuclear hormone signaling (LXR/RXR Activation) was predicted to be activated in male brain and blood (Fig 2C).

## CIE-responsive gene networks conserved between blood and brain

We identified conserved gene coexpression modules in blood and brain. We built gene coexpression networks for each tissue individually and determined whether the different tissues had similar modules, i.e., were comprised of the same genes (if so, these modules were said to be conserved between blood and brain and are referred to as blood-brain modules).

To link the blood-brain modules to CIE exposure, we identified modules enriched with DEGs or with eigengenes correlated with alcohol preference or consumption levels in the final drinking test, or group (CIE and Air). To gain functional insight into the blood-brain modules, we performed an IPA Core Analysis on the genes within the blood-brain modules (S4 and S5 Tables). IPA analysis included predicted upstream regulators which are transcription factors, chemical compounds, microRNAs, or other regulators that might explain the observed changes in gene expression. If a predicted upstream regulator of a blood-brain module was also a member of the blood-brain module, this might indicate a particularly important role for that gene in regulating the transcriptional response to CIE-induced alcohol dependence. We highlight these genes in the following sections as we describe each blood-brain module identified in the network analyses.

We found three main groups of alcohol dependence-related blood-brain modules for female mice and four for male mice (boxes in Fig 3). The modules conserved between blood and brain were similar in males and females as revealed by the categories that emerged from IPA Core Analysis of the genes within the blood-brain modules. For example, there was a "cell-cell signaling" module (Fig 3) (e.g., endocannabinoid signaling, GABA and glutamate receptor signaling, synaptogenesis), an "immune response" module (Fig 3) (e.g., antigen presentation, communication between innate and adaptive immune systems, JAK/STAT signaling), and a "protein processing / mitochondrial function" module (Fig 3) (e.g., ubiquitination, unfolded protein responses, oxidative phosphorylation) for both males and females. Male mice also exhibited a "fatty acid metabolism / peroxisome proliferator activated receptor (PPAR)" blood-brain module (Fig 3).

## Network analysis in female mice

The "cell-cell signaling" module was particularly highly conserved across blood and brain in female mice (Fig 3; edge thickness reflects degree of overlap). The genes in these modules were mostly up-regulated in blood and brain after CIE. Interestingly, this group of modules was enriched with genes already associated with alcohol-related behavior [9, 10, 40] (via mutant

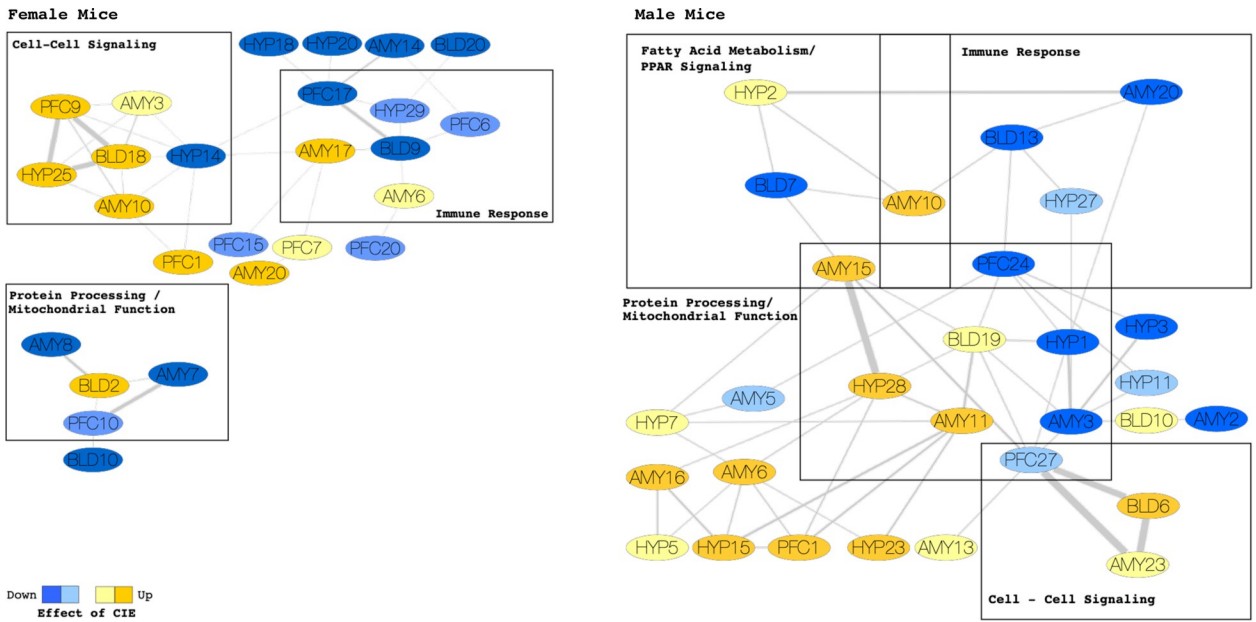

**Fig 3. Blood/Brain Gene Coexpression Modules Affected by CIE-induced Alcohol Dependence.** A meta-network of overlapping gene coexpression modules in blood (BLD) and brain [prefrontal cortex (PFC), amygdala (AMY), hypothalamus (HYP)] in female (left) and male (right) C57BL/6J mice. Each node represents a module of coexpressed genes. Nodes are labeled with tissue type and a module number. An edge between two nodes indicates a significant overlap of genes between two modules of different tissues. Thickness of connecting edges is proportional to the significance of the overlap. Module colors represent the direction and magnitude of regulation of CIE-induced alcohol dependence based on the significance of the enrichment with differentially expressed genes (yellow, upregulation; blue, downregulation in CIE mice; intense colors, p < 0.00001; light colors, p < 0.05). Only modules affected by CIE are shown. An overlap of a blood module and at least two modules from different brain regions indicates a cluster of highly conserved coexpression modules regulated by CIE in blood and brain (termed blood-brain modules in the text; represented by rectangular boxes in the figure). All overlapping modules within a blood-brain cluster were overrepresented with genes from a major biological category. These were the same for male and female mice apart from a Fatty Acid Metabolism and PPAR Signaling cluster unique to male mice. The top 5 enriched pathways and predicted upstream regulators from the IPA analysis for each of the broad categories (rectangular boxes) are reported in the text, but examples include GABA and Glutamate Receptor Signaling for the Cell-Cell Signaling cluster, Inflammasome Pathway and Th1 Pathway for the Immune Response cluster, Protein Ubiquitination Pathway and Oxidative Phosphorylation for the Protein Processing / Mitochondrial Function cluster, and Fatty Acid Oxidation and PPARA for the Fatty Acid Metabolism and PPAR Signaling cluster. The overlap between blood and brain modules was particularly strong for the Cell-Cell Signaling clusters in both male and female coexpression networks. The Cell-Cell Signaling cluster also contained genes known to play a role in alcohol-related behaviors in rodents (e.g., *Bdnf*, *Npy*, *Gabra1*, and *Pdyn*). Network visualization was performed using Cytoscape version 3.8.2.

mouse studies) including: *Adcy1*, *Grin2a*, *Grm5*, *Hrh3*, *Npy1r*, *Adcy5*, *Cacna1b*, *Chrna4*, *Faah*, *Gabrb2*, *Gabrd*, *Grm4*, *Hras*, *Prkar2b*, *Prkce*, *Bdnf*, *Cckbr*, *Cnr1*, *Gabra1*, *Pdyn*, and *Prkar1b*. The top five predicted upstream regulators for this preserved module were *Rest*, *Hdac4*, *Snca*, *Fmr1*, *Htt*. Several of the predicted upstream regulators were also members of the module, suggesting these genes may be particularly important regulators of the blood-brain "cell-cell signaling" module in CIE-induced alcohol dependence (in decreasing order of significance): *Bdnf*, *Mapt*, *App*, *Tshz3*, *Slc30a3*, *Shank3*, *Nfasc*, *Kcnk9*, *Slitrk5*, *Fezf2*, *Gabra1*, *Ntrk3*, *Fbxo2*, *Mapk8ip1*, *Lhx2*, *Dmd*, *Slc9a6*, *Agrn*, *Gabbr2*, *Htr2a*, *Akap5*, *Dpp10*, *Bhlhe22*, *Kcnd3*, *Scn1b*, *Nptx1*, *Dlg3*, *Baiap2*, *Pdyn*, *Cacnb4*, *Elavl4*, *Grm3*, *Kif1b*, *Dab1*, *Dnm1*. Of these, *Bdnf*, *Gabra1*, and *Pdyn* are genes already associated with alcohol-related behavior.

The "immune response" module (Fig 3) was down-regulated in blood, PFC, and HYP and up-regulated in AMY after CIE. The top five predicted upstream regulators were: *Cst5*, mir-17, betulinic acid, *Mmp3*, miR-17-5p (and other miRNAs w/seed AAAGUGC). Several of the predicted upstream regulators were also members of the module, suggesting these genes may be particularly important regulators of the blood-brain "immune response" module in CIE-

induced alcohol dependence (in decreasing order of significance): *Nup107*, *Rbm5*, *Tlr4*, *Ly96*, *Sf3b1*, *Aim2*, *Tnrc6a*, *Abca1*.

The "protein processing / mitochondrial function" module (Fig 3) was up-regulated in blood and down-regulated in PFC and AMY after CIE. There was not a corresponding module in the HYP. The top five predicted upstream regulators were: *Abcb6*, *Hipk2*, enterotoxin B, Torin1, and *Fancd2*. Several of the predicted upstream regulators were also module members suggesting these genes may be particularly important regulators of the blood-brain response to CIE-induced alcohol dependence related to protein processing and mitochondrial function / oxidative phosphorylation (in decreasing order of significance): *Bnip3l*, *Gpx1*, *Irf7*, *Adipor1*, *Uros*, *Mafg*.

## Network analysis in male mice

Similar to female mice, the overlap between blood and brain modules was strongest for the "cell-cell signaling" blood-brain module. The genes in the "cell-cell signaling module" in males (Fig 3) tended to be up-regulated in blood and AMY, and down-regulated in PFC after CIE. There was not a corresponding module in the HYP. Similar to female mice, the blood module was enriched with genes already associated with alcohol-related behavior including: *Adcy1*, *Adcy5*, *Adora2a*, *Bdnf*, *Cckbr*, *Cnr1*, *Gabra2*, *Gabra5*, *Gabrd*, *Grin2a*, *Homer2*, *Hrh3*, *Kcnj6*, *Npy*, *Prkce*, *Prkg2*. The top five predicted upstream regulators were *Jak1/2*, *Snca*, *Hdac4*, *Fmr1*, *Htt*. Several of the predicted regulators were also in the cell-cell signaling module (in decreasing order of significance): *Bdnf*, *Tshz3*, *Slc30a3*, *Fezf2*, *Homer2*, *Bhlhe22*, *Prmt8*, *Grm2*, *Fkbp1b*, *Baiap2*, *NpaS3*, *Grm3*, *Nfib*, *Chrm1*, *Gabbr2*, *Npy*. Of these, *Bdnf*, *Homer2*, and *Npy* are genes already associated with alcohol-related behavior.

The "immune response" module in males (Fig 3) was down-regulated in blood, PFC, HYP, and both up- and down-regulated in AMY after CIE. The top five predicted upstream regulators were: miR-124-3p (and other miRNAs with seed AAGGCAC), lipopolysaccharide, *Pgr*, *Rel*, *Btnl2*. Several of the upstream regulators were also in the immune response module (in decreasing order of significance): *Csf1r*, *Stat3*, *Vcan*, *Spi1*, and *Notch1*.

The "protein processing / mitochondrial function" module in males (Fig 3) was up-regulated in blood and both up- and down-regulated in brain after CIE. The top five predicted upstream regulators were: *Abcb6*, 1,2-dithiol-3-thione, *Klf1*, *Rictor*, *St1926*. Several upstream regulators were also module members (in decreasing order of significance): *Bnip3l*, *Fth1*, *Cdc25b*, *Grp*, *Mrpl12*, *Ola1*, *Tpm1*, *Fancd2*, *Ctsb*, *E2f2*, *Uba1*, *E2f3*, *Cul1*, *Rb1cc1*, *Fzr1*, *Sub1*, *Stx2*, *Rb1*, *Gadd45a*, *Gsk3a*.

The "fatty acid metabolism / peroxisome proliferator activated receptor (PPAR)" module in males (Fig 3) was down-regulated in blood and up-regulated in AMY and HYP. There was not a corresponding module in PFC. The top five predicted upstream regulators were: pirinixic acid, bezafibrate, methotrexate, *Ppara*, *Ehhadh*. Three of the upstream regulators were also in the fatty acid metabolism / PPAR" module: *Adipor2*, *Acox1*, *Nrg4*.

## Correlation of blood and brain gene expression levels

**Between-subjects analysis.** We studied the preservation of mean gene expression levels of the genes between brain and blood (irrespective of treatment). The pairwise scatterplots in S1 Fig related mean expression values in the three brain regions to mean expression values in blood. We found significant correlations (rho range males: [0.67, 0.67], rho range females: [0.50,0.51]) between mean expression in brain and mean expression in blood (S1 Fig). The correlation between blood and brain expression levels was notably higher in males relative to females (S6 Table).

**Within-subjects analysis.**   To determine the genes with expression levels that are correlated between brain and blood, we calculated the within-subject correlation between gene levels in blood and brain. Expression levels of hundreds of genes were significantly correlated between blood and brain even after correcting for multiple comparisons (Tables 1 and S7). Table 1 shows the top ten genes correlated between blood and brain for each brain region. To gain insight into the cellular specificity of the correlated genes, we determined whether any cell type-specific genes were over-represented in each of the gene sets. The enriched cell types are noted in Table 1. The genes correlated between amygdala and blood in both males and females were enriched with microglial markers, the majority of which were negatively correlated between these tissues. The genes correlated between PFC and blood in both males and females were enriched with endothelial markers, with most negatively correlated between blood and PFC. Additionally, the genes correlated between PFC and blood in males were enriched with microglial markers and the genes correlated between amygdala and blood in males were enriched with T cell markers; all were negatively correlated between blood and PFC.

To gain functional insight into the genes correlated between blood and brain, we performed pathway enrichment analysis on each gene set (S8 Table). There was little overlap between the enriched pathways across the correlated gene sets between blood and the different brain regions. For example, cholesterol biosynthesis and ethanol degradation pathways were prominent in genes correlated between female blood and PFC (but no other brain region). However, there were some overlapping pathways between blood and multiple brain regions. Glucocorticoid receptor signaling and regulation pathways were among the most commonly enriched pathways, found in the genes correlated between blood and all 3 brain regions for females and between blood and hypothalamus for males. Interferon signaling and antiviral response pathways were commonly enriched in the genes correlated between blood and hypothalamus and blood and PFC in both sexes. MIF Regulation of Innate Immunity was common to male hypothalamus and female amygdala. IL-17 production and signaling pathways were common to female amygdala and hypothalamus. DNA methylation and transcriptional regulation pathways were common to female amygdala and PFC, and male amygdala. Pathways related to DNA damage responses were common to in the correlated gene sets between blood and multiple brain regions as well (e.g., base excision repair for male amygdala and female hypothalamus, nucleotide excision repair for female amygdala, and role of BRCA1 in DNA damage response for male hypothalamus). Methionine Degradation was common to the PFC of both sexes and female hypothalamus.

Some of the correlated genes were also differentially expressed between CIE and Air mice in both blood and brain (Tables 2–4). For males, *Hsp1a* and *Hsp1b* were correlated between

**Table 1.  Genes Correlated Between Blood and Brain.**

| | Females | | | Males | | |
|---|---|---|---|---|---|---|
| | Number of Corr Genes | Enriched Cell Type | Top 10 Corr Genes | Number of Corr Genes | Enriched Cell Type | Top 10 Corr Genes |
| **Amygdala** | 542 | mic | Pacrg, Hgf, Pofut1, Helb, Tmem131, Hpgd, Card19, Clec12a, Setx, Rbm12b2 | 707 | mic, T cells | Zfp521, Gm4799, Ghr, Zfp385a, Kirrel3, Ceacam1, Gm32647, Dcp1a, Zfp217, Sema3f |
| **PFC** | 634 | endo | Smarcc2, Kcnk3, Smim10l1, Agps, Rdh11, Hivep2, Raver2, Gm42984, Cbfa2t3, Cep57l1 | 656 | endo, mic | Lurap1l, Tenm4, Mdh1, Elf4, Pyurf, Sod1, Ccdc166, Ube3b, Slc36a1, Snx20 |
| **Hypothalamus** | 649 | None | Dglucy, Hsp90aa1, Lnpep, Btaf1, Hmox2, Usp6nl, R3hcc1l, Uap1l1, Jade1, Taf13 | 686 | None | Hdac11, Arf6, Sdk2, Scamp1, Gm46515, Fgd1, Wwp2, Zscan18, Fam13c, Espn |

Mic = microglia, endo = endothelial cells, PFC = prefrontal cortex

**Table 2. Genes Differentially Expressed between CIE and Air C57Bl/6J Mice in Amygdala with Expression Levels that are also Correlated between Blood and Amygdala.**

| Male | | Female | |
|---|---|---|---|
| Gene Symbol | Gene Name | Gene Symbol | Gene Name |
| Acaa2 | Acetyl-CoA acyltransferase 2 | Ccl5 | C-C Motif Chemokine Ligand 5 |
| Ccp110 | Centriolar Coiled-Coil Protein 110 | Cdc42ep2 | CDC42 Effector Protein 2 |
| Dennd4b | DENN/MADD domain containing 4B | Dusp10 | Dual Specificity Phosphatase 10 |
| Galm | Galactose Mutarotase | Gabpb1 | GA Binding Protein Transcription Factor Subunit Beta 1 |
| Hspa1a | Heat Shock Protein Family A (Hsp70) Member 1A | Setx | Senataxin |
| Hspa1b | Heat Shock Protein Family A (Hsp70) Member 1B | Trim25 | Tripartite Motif Containing 25 |
| Hspa5 | Heat Shock Protein Family A (Hsp70) Member 5 | Vipr1 | Vasoactive Intestinal Peptide Receptor 1 |
| Mgst1 | Microsomal Glutathione S-Transferase 1 | | |
| Pou6f1 | POU Class 6 Homeobox 1 | | |
| Pygl | Glycogen Phosphorylase L | | |
| S100a6 | S100 Calcium Binding Protein A6 | | |
| Vav1 | Vav Guanine Nucleotide Exchange Factor 1 | | |

blood and each brain area and differentially expressed in blood and all brain regions, as was *Pygl* (except for hypothalamus) (Fig 4). Blood and brain levels of *Hspa1a* and *Hspa1b* tended to be negatively correlated with alcohol consumption in the last 2BC drinking test, while *Pygl* had weak correlations with alcohol intake (S2 Fig). For females, *Ccl5* was correlated between blood and each brain area and differentially expressed in blood and all brain regions, as was *Vipr1* (except for hypothalamus) (Fig 4). Blood expression levels of *Ccl5* were positively correlated with alcohol intake, while blood expression levels of *Vipr1* were negatively correlated with alcohol intake (S2 Fig). We created an R Shiny app where interested readers can see the correlation between voluntary ethanol intake levels and the expression level of any gene of interest (https://lauraferguson.shinyapps.io/blood_brain/).

## Whole blood gene expression signatures can distinguish CIE and Air subjects

We determined whether CIE status could be predicted by blood gene expression profiles using three different classification techniques: logistic regression (LR) with elastic net regularization,

**Table 3. Genes Differentially Expressed between CIE and Air C57Bl/6J Mice in Hypothalamus with Expression Levels that are also Correlated between Blood and Hypothalamus.**

| Male | | Female | |
|---|---|---|---|
| Gene Symbol | Gene Name | Gene Symbol | Gene Name |
| Chsy1 | Chondroitin Sulfate Synthase 1 | Aacs | Acetoacetyl-CoA Synthetase |
| Clec2d | C-Type Lectin Domain Family 2 Member D | Bcl2l11 | BCL2 Like 11 |
| Dnajb11 | DnaJ Heat Shock Protein Family (Hsp40) Member B11 | Caln1 | Calneuron 1 |
| Fam13c | Family With Sequence Similarity 13 Member C | Card6 | Caspase Recruitment Domain Family Member 6 |
| Gm46515 | predicted gene, 46515 (lncRNA) | Ccl5 | C-C Motif Chemokine Ligand 5 |
| Hspa1a | Heat Shock Protein Family A (Hsp70) Member 1A | Gpm6b | Glycoprotein M6B |
| Hspa1b | Heat Shock Protein Family A (Hsp70) Member 1B | Gria1 | Glutamate Ionotropic Receptor AMPA Type Subunit 1 |
| Kdsr | 3-Ketodihydrosphingosine Reductase | Smyd1 | SET And MYND Domain Containing 1 |
| Parp9 | Poly(ADP-Ribose) Polymerase Family Member 9 | Vrk2 | VRK Serine/Threonine Kinase |
| Spred2 | Sprouty Related EVH1 Domain Containing 2 | Zcchc3 | Zinc Finger CCHC-Type Containing 3 |
| Sptssa | Serine Palmitoyltransferase Small Subunit A | | |

**Table 4. Genes Differentially Expressed between CIE and Air C57Bl/6J Mice in Blood and PFC with Expression Levels that are also Correlated between Blood and PFC.**

| Male | | Female | |
|---|---|---|---|
| Gene Symbol | Gene Name | Gene Symbol | Gene Name |
| Ahsa2 | Activator of heat shock 90kDa protein ATPase homolog 2 | 4930549G23Rik | RIKEN cDNA 4930549G23 gene (lncRNA) |
| Cebpb | CCAAT Enhancer Binding Protein Beta | Atp6v0e2 | ATPase H+ Transporting V0 Subunit E2 |
| Dnajb1 | DnaJ Heat Shock Protein Family (Hsp40) Member B1 | Btbd8 | BTB Domain Containing 8 |
| Dnlz | DNL-Type Zinc Finger | Ccl5 | C-C Motif Chemokine Ligand 5 |
| Far1 | Fatty Acyl-CoA Reductase 1 | Efcab5 | EF-Hand Calcium Binding Domain 5 |
| Fkbp5 | FKBP Prolyl Isomerase 5 | Hnrnpf | Heterogeneous Nuclear Ribonucleoprotein F |
| Hspa1a | Heat Shock Protein Family A (Hsp70) Member 1A | Vipr1 | Vasoactive Intestinal Peptide Receptor 1 |
| Hspa1b | Heat Shock Protein Family A (Hsp70) Member 1B | | |
| Insig1 | Insulin Induced Gene 1 | | |
| Naaladl2 | N-Acetylated Alpha-Linked Acidic Dipeptidase Like 2 | | |
| Pygl | Glycogen Phosphorylase L | | |
| Zfp329 | Zinc finger protein 329 | | |

random forest (RF), and partial least squares discriminant analysis (PLSDA). Ideally, in addition to being highly accurate at the classification task, a good model would also be interpretable. We chose these three techniques because they included measures of variable importance which enabled interpretation of the models. Furthermore, because the dataset is small, we wanted to use noncomplex models to avoid overfitting. We used all genes to train the classifiers. Some of the coefficients in the regularized LR model go to zero and those features fall out of the model. Some features in the RF model have zero Mean Decrease in Gini metric and are not included in the final model. In this way the regularized LR and RF methods include embedded feature selection. The PLSDA model includes variable importance measures but does not include built-in feature selection and uses all genes for the model.

For female subjects, the logistic regression model performed best, and was highly accurate at identifying CIE subjects from controls (AUC 90.1%) using only 32 genes (*Gatad2a*, *ENSMUSG00000096544.2*, *Plgrkt*, *E030030I06Rik*, *ENSMUSG00000085633.1*, *Ccdc85b*, *Hsd11b1*, *Gzma*, *Utp14a*, *Zfp472*, *Foxq1*, *Cope*, *Saxo2*, *Ankle1*, *Med10*, *Gm4841*, *Ubn1*, *Sephs1*, *Smarca4*, *Phf6*, *Eri1*, *Aldh7a1*, *Il4i1*, *Pigz*, *Apol11a*, *Chchd6*, *Lrrc8c*, *Mlec*, *Cenps*, *Cct5*, *Apba2*, *ENSMUSG00000109006.2*) (Fig 5A). The partial least squares and random forest techniques were also able to identify CIE subjects from controls with a high degree of accuracy (PLSDA AUC was 80.8% and random forest AUC was 79.2%) (Fig 5A). Random forest required 807 genes to achieve its high performance (Fig 5A).

For male subjects, partial least squares performed best at identifying CIE subjects from controls (AUC 80.5%) (Fig 5C). Logistic regression also performed well (AUC 75.9%) and required only 2 genes (*Plet1* and *Hspa5*) (Fig 5C). The random forest model employed 382 genes. It was not able to distinguish between CIE and Air mice very well (AUC 58.2%) (Fig 5C).

To determine which genes were the most important for classifying the subjects as CIE and Air mice, we examined the variable importance measures from random forest, partial least squares discriminant analysis, and logistic regression. There was little overlap between the top important genes between the different models. For females *Gzma* was common to logistic regression and partial least squares discriminant analysis, *Apol11a* was common to random forest and partial least squares discriminant analysis (*Apol11a* was also one of the 32 genes with non-zero coefficients in the logistic regression model), and *Gatad2a*, *Hsd11b1*, and

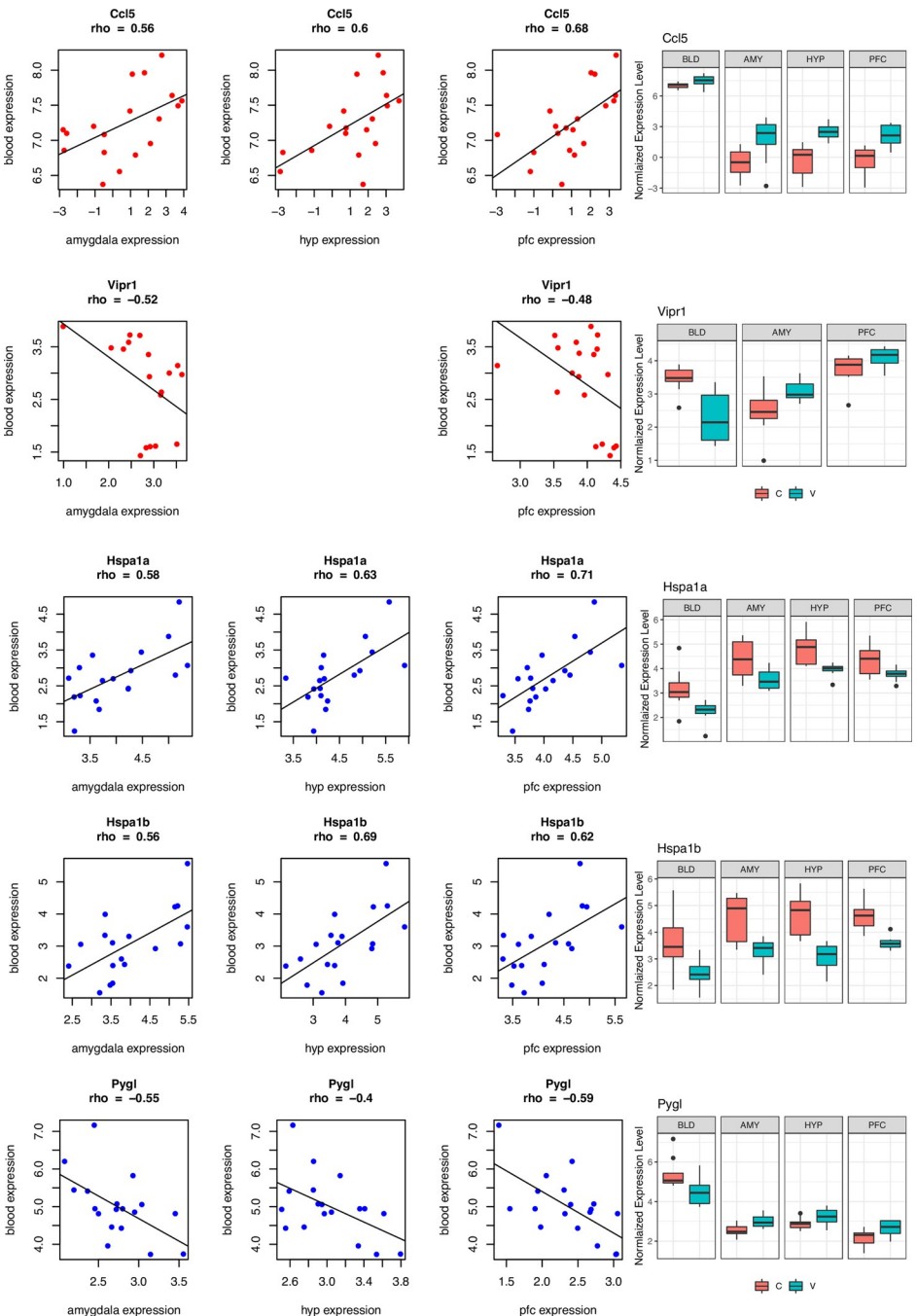

**Fig 4. Genes with Expression Levels that are Correlated between Blood and Brain in C57BL/6J Mice.** The scatterplots show the relationship between gene expression levels in blood (y-axis) and each brain area (x-axis). Each point in the scatterplot is a subject. Some correlated genes were also differentially expressed between CIE (V = ethanol vapor) and Air (C = control) mice in both blood and brain, which could make them ideal biomarker candidates. For females, *Ccl5* was correlated and differentially expressed in blood and all brain regions. For males, *Hsp1a* and *Hsp1b* were correlated and differentially expressed in blood and all brain regions. The correlation coefficient is shown under the gene name. The boxplots show the normalized gene expression levels by group.

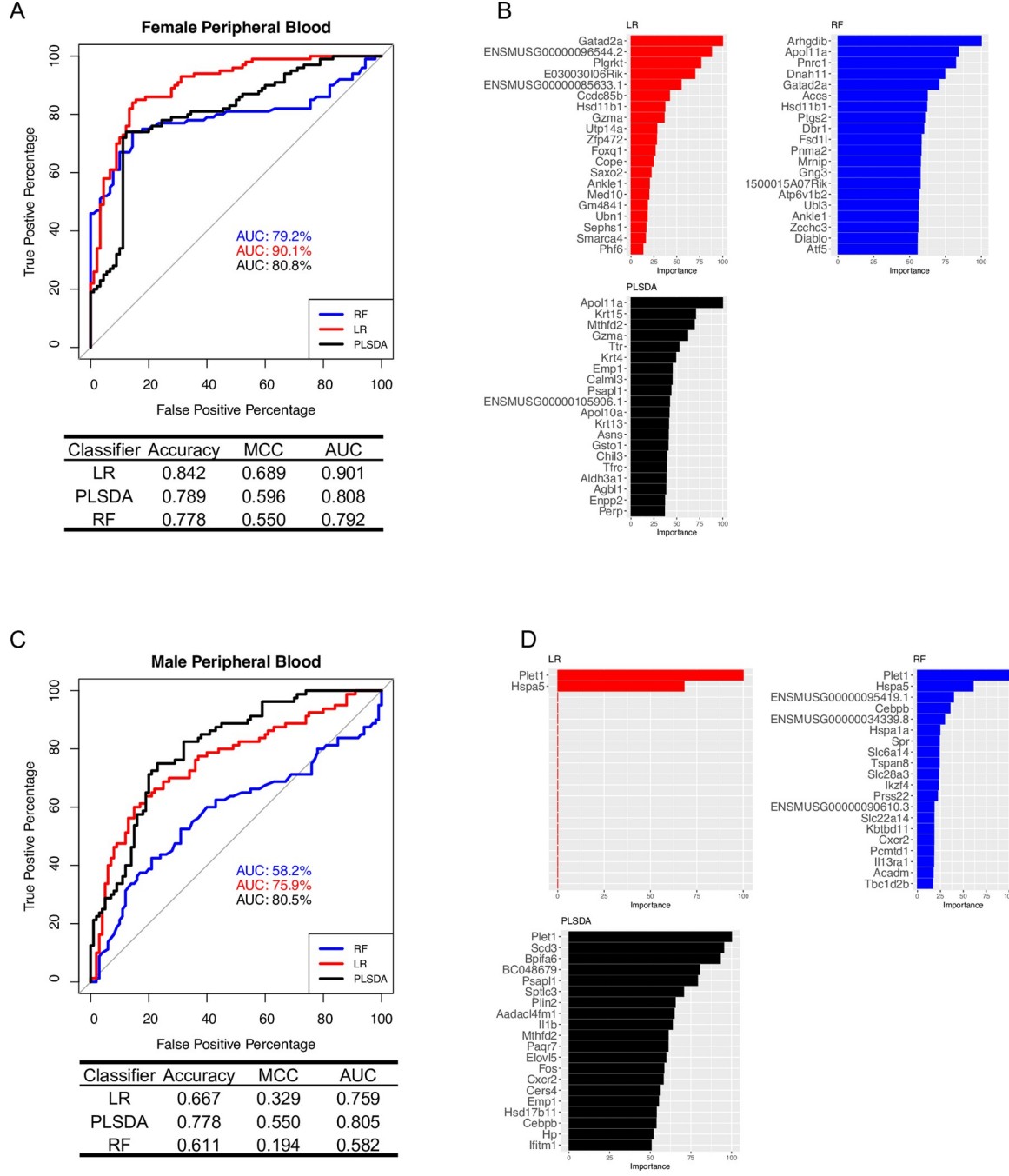

**Fig 5. Classification Performance Using Blood Gene Expression.** Machine learning classifiers were trained using peripheral whole blood gene expression data to predict alcohol dependence status. Receiver Operating Characteristic (ROC) curves are shown for the classification techniques indicated by the line type in the legend for (A) female and (C) male C57BL/6J mice. The ROC curve shows the relationship between the True Positive Rate (y-axis) and the False Positive Rate (x-axis) resulting from a set of binary classification tests based on each possible decision threshold value. The area under the curve (AUC) for each classifier is displayed in the graph in a corresponding color. The diagonal represents chance levels which corresponds to an AUC of 0.50. A table below the ROC curve shows different performance metrics of the classifiers. The top 20 important genes for the classification task are shown for (B) females and (D) males for the separate models. RF = Random Forest, LR = Logistic Regression, PLSDA = Partial Least Squares Discriminant Analysis, MCC = Matthews Correlation Coefficient.

*Ankle1* were common the logistic regression and random forest (Fig 5B). For males, *Plet1* was common to all three classifiers, *Hspa5* was common to logistic regression and random forest, and *Cebpb* and *Cxcr2* were common to random forest and partial least squares discriminant analysis (Fig 5D).

## Discussion

Routine blood testing has long been a part of medical care. Genomic profiles in blood could be used to provide data-driven diagnosis of AUD, stratify the heterogeneous AUD patient population for clinical trials, select optimal therapy, determine whether a patient has AUD risk, and monitor the efficacy of the patient's medications. Here we performed a well-controlled animal model study to conduct whole genome profiling of brain and blood with ethanol treatment. An important question this study addressed was whether blood gene expression signatures could predict CIE and Air mice. The highly predictive performance of the classifiers suggests that there is valuable diagnostic information in blood gene expression for CIE-induced alcohol dependence, even one week since the last alcohol exposure. Different features were selected by the different classification methods, which is consistent with the literature [41] and expected because the methods employ different techniques for the classification task and the features in gene expression datasets are highly correlated. These results confirm that good performance in the classification task can be achieved using different sets of features. That (and the dynamic nature of gene expression) might make it challenging to select a consistent panel of biomarkers and it will likely be necessary to incorporate other sources of information into a diagnostics screen. There were however several features ranked as highly important for achieving good performance in multiple models. This included *Gzma*, *Apol11a*, *Gatad2a*, *Hsd11b1*, and *Ankle1* for females and *Plet1*, *Hspa5*, *Cebpb*, and *Cxcr2* for males. *Cebpb* and *Cxcr2* are especially interesting considering the importance of immune molecules in rewponses to alcohol (discussed below). However, immune-responsive molecules may not be ideal biomarkers given that immune responses can be initiated by infections or other environmental perturbations.

AUROC measures the quality of the model's predictions irrespective of what classification threshold is chosen. The classification threshold strikes a balance between false positive and false negative rates, and the optimal choice would likely depend on the setting and purpose (e.g. primary care screening for problematic drinking versus criminal justice settings) [42]. Moreover, the predictive value of these classifiers will depend on the prevalence of the disorder in the population tested. A population with low prevalence of the disorder (for example, in primary care screening for AUD) results in an increase in false positive tests, whereas a higher prevalence rate (e.g., in an addiction clinic) yields more false-negative tests [42]. Here in the controlled experimental conditions the prevalence was 50/50. Thus, our findings represent important first steps in identifying novel biomarkers for AUD, but more research (including human validation) will be required before genes selected as discriminatory features could become viable biomarkers for AUD.

The mice in this study have been exposed to weeks of high dose ethanol vapor and would likely represent severe cases of AUD. These cases are usually clinically evident by patient history and physical exam and would not necessitate a molecular diagnostic tool. However, social stigma surrounding alcoholism (like most psychiatric disorders) remains a barrier to treatment. An objective molecular diagnostic tool (e.g., a "blood test") would potentially mitigate stigma and encourage patients to seek diagnosis and treatment for this disorder. Additionally, blood expression profiles might detect milder forms of AUD or those at high-risk for developing AUD. The former could be investigated in future studies that include an additional

ethanol-naïve control group or a larger number of animals to allow analysis of low versus high drinkers. The latter could be addressed in future studies by profiling the blood of ethanol-naïve genetic rodent models of AUD-risk traits (e.g., high alcohol preference or consumption, propensity for binge-like drinking), or by analyzing the blood transcriptome before and after CIE treatment and correlating gene expression changes with the amount of drinking escalation within subjects. The blood transcriptome varies between people, but it is relatively stable for an individual across time [43] which is encouraging and suggests that repeated blood sampling could be used for personalized medicine approaches. Studies that analyze gene expression profiles in blood sampled at multiple time points throughout the addiction cycle (including before alcohol exposure) will be critical in exploring the clinical utility of blood gene expression profiles for AUD.

Another question that this study examined was whether expression levels of the genes between brain and blood are preserved. While some studies have shown significant but weak correlations between blood and brain (e.g., [44]), we found that blood and brain average gene expression levels were highly correlated. The difference is likely attributable to the fact that we compared blood and brain samples from the same subjects instead of comparing blood and brain samples from different individuals. Strikingly, our within-subjects design revealed that the expression levels of hundreds of genes were significantly correlated between blood and brain. This was irrespective of treatment, which suggests that the blood might be useful for gaining insight into brain functioning in general. Indeed, there are many compelling examples from the literature that support the idea that whole blood transcriptomes can be informative for a number of brain diseases [45–47].

The transcriptional response to alcohol dependence in brain showed stronger conservation at the level of coexpression than at the level of individual genes. We identified an "immune response" network similarly perturbed in blood and brain after CIE. Immune functions also dominated the IPA Core Analysis results for blood and brain DEGs. This finding adds to the importance of "neuroimmune" signaling in alcohol dependence and reflects the key role white blood cells play in immune responses as they are the source for most of whole blood mRNA. The relationship between alcohol and the immune system has been the subject of intensive research, and much insight has emerged in the last 15 years pointing to a bi-directional relationship between alcohol consumption and immune signaling molecules, whereby alcohol ingestion increases peripheral and central cytokine levels. Conversely, manipulation of cytokines and other immune signaling molecules can increase alcohol consumption and craving [48–52], leading to further increases in cytokine levels and emergence of an out-of-control, positive-feedback cycle. This cycle demonstrates the systemic nature of AUD and the involvement of peripheral and central immune signaling in AUD pathophysiology [7].

The details of peripheral-central immune signaling crosstalk involved in alcohol dependence remain to be fully understood. One way alcohol triggers inflammatory responses is via reactive oxygen species produced during ethanol metabolism, which can occur both peripherally and centrally. Alcohol also increases intestinal permeability ("leaky gut"), which permits gut-derived bacterial products to enter the circulation where they are recognized by immune cells in blood or target organs, resulting in the release of pro-inflammatory cytokines [53]. Peripheral cytokines can trigger central "immune" responses via vagal afferents or by crossing the blood brain barrier to enter the brain. Although these are plausible mechanisms, it remains to be shown the exact peripheral-central communication that must occur, which likely involves multiple mechanisms. Our IPA analysis predicted an increase in leukocyte migration and number in brain and a decrease in blood. The cell type enrichment analysis of the DEGs revealed an up-regulation of leukocyte genes in brain and a down-regulation in blood. Also, the genes correlated between blood and brain were enriched with endothelial, microglia, and

leukocyte genes and tended to be negatively correlated between blood and brain. Taken together, these findings suggest that white blood cells could be recruited into the CNS during alcohol dependence. Future studies are planned to study this novel hypothesis as an unexplored mechanism of peripheral-central immune crosstalk in alcohol dependence.

One of the most surprising findings of this study was that the most highly conserved coexpression networks for both sexes were related to cell-cell signaling and included such "brain-related" categories as glutamate and GABA receptor signaling. The importance of these systems in the CNS has been long-known for alcohol dependence (reviewed in [54]), but our results suggest that *blood* GABA and glutamatergic signaling are also perturbed in alcohol dependence. This has been partially validated by previous studies that found that GABA serum levels are lower and glutamate serum levels are higher in alcoholic patients compared with non-dependent controls during alcohol withdrawal [55, 56]. GABA regulates the secretion of cytokines from PBMCs in a concentration-dependent manner [57] providing another novel potential link between alcohol and immune regulation suggested by this study. Moreover, glutamate serum levels upon hospital admission are predictive of developing an alcohol withdrawal syndrome 12 h later [55]. Glutamate serum levels are correlated with brain levels [58] and are predictive of other brain diseases such as multiple sclerosis [59], schizophrenia [60], and autism [61]. The cell-cell signaling blood-brain module also contained numerous genes that modulate alcohol consumption (as determined by mutant mouse studies). Of these *Adcy1*, *Bdnf*, *Cckbr*, *Cnr1*, *Gabra5*, *Gabrd*, *Grin2a*, and *Hrh3* were common to both sexes and *Bdnf* was also a hub in the Graphical Summary for the female blood DEGs in Fig 2C. Because most of the mutant mouse studies were global knockouts and these genes are ubiquitously expressed, this result calls into question the brain-specificity of the knockout findings. Perhaps there is a larger peripheral component to the causal effects of these genes on alcohol consummatory behavior than previously appreciated.

We identified a "protein processing / mitochondrial function" blood-brain module in both sexes. This included genes involved in protein ubiquitination (including ubiquitin B, numerous ubiquitin conjugating enzymes, ubiquitin specific peptidase, heat shock factors), mitochondrial dysfunction (including *Pink1*, *Fis1*, and many NADH:ubiquinone oxidoreductase subunits), and ethanol degradation (*Cat*, *Aldh1a1*). It is possible that this conserved coexpression network represents a response to ethanol-induced cellular stress. For males, genes coding for the Heat Shock Protein Family A (Hsp70) subunits, *Hspa1a* and *Hspa1b*, were upregulated and correlated between blood and all three brain areas tested. Another Hsp70 member, *Hspa5*, was downregulated in all male tissues, correlated between blood and brain, and also an important gene for the classification tasks as discussed above. *Hspa1a*, *Hspa1b*, and *Hspa5* code for potent anti-inflammatory proteins that can initiate protective responses to stress. It has been postulated that the cardioprotective effect of alcohol consumption is due in part to increased intracellular HSPA1A [62]. Acute alcohol exposure induces *Hspa1a* in human monocytes and is required for inhibition of TLR4/MyD88 (but not TLR4/TRIF) signaling in macrophages [63]. *Hspa1a* and *Hspa1b* transcripts are also dysregulated after ethanol exposure in rodent in total brain homogenates [64–66], astrocytes [67, 68], and microglia [68]. Depending on the time point at which gene expression was assayed and perhaps other differences between the protocols employed, these studies have shown increases or decreases in their transcript abundance. Patients with alcoholic hepatitis exhibit lower gene expression levels of HSPA1A in liver compared with healthy controls [69]. *In utero* exposure to ethanol increases *Hspa1a* in cortical tissue from humans and mice [70]. Hspa5 is linked to alcohol consumption and withdrawal in rodents [71–75] and induced by chronic alcohol in cell lines [76]. In addition to their roles as protein chaperones, *Hspa1a* and *Hspa1b* are known splicing factors [77]. Another splicing factor in the Hsp70 family, Hspa6, is drastically increased in postmortem frontal

cortex and amygdala samples from AUD patients and thought to be important for the observed genome-wide changes in splicing observed in these brain regions with AUD [77]. From these examples, it is clear that Hsp70 subunits are responsive to ethanol in both brain and blood, and our analysis suggests that levels in blood are reflective of those in brain and might be useful biomarkers of alcohol dependence.

In females, transcripts for C-C Motif Chemokine Ligand 5 (*Ccl5*) were increased with CIE and levels of expression were correlated between blood and brain in all three brain regions. *Ccl5* transcripts are also increased in total homogenate and astrocytes in male mouse cortex after ethanol vapor exposure [68]. CCL5 transcripts are reduced in the central amygdala of human alcoholic subjects compared with controls [78] as well as in the brains of ethanol-naïve rodent lines that drink high amounts of alcohol [79]. In females, transcripts for the Vasoactive Intestinal Peptide Receptor 1, *Vipr1*, were reduced in blood and increased in the amygdala and PFC, with *Vipr1* expression levels negatively correlated between blood and these brain regions. VIPR1 transcripts are also increased in the frontal cortex of human alcoholic subjects compared with controls [78] and reduced in microglia from mouse PFC after CIE [68]. There is a single nucleotide polymorphism in *VIPR1* that is associated with bipolar disorder [80]. *VIPR1* is also a hub gene that acts as a prognosis and progression biomarker for hepatocellular carcinoma [81].

Consistent with previous studies of female mice and rats [82–87], we found a less robust escalation of voluntary ethanol intake after CIE in females compared with males. This finding could be partially due to a ceiling effect (the female mice begin drinking at levels about three times higher than males before CIE). Nevertheless, our study revealed that female mice display a strong molecular phenotype after CIE. Notably, the transcriptome signature in female peripheral blood was able to discriminate between CIE and Air female mice with very high accuracy, even more-so than in male mice. Therefore, although female mice did not exhibit a robust escalation in voluntary drinking after CIE, they showed transcriptional changes that were particularly strong. Network analysis and IPA analysis showed that females and males had similar blood-brain coexpression modules affected by alcohol dependence, suggesting that this molecular phenotype could be comparable between the sexes, at least at the level of gene networks. One exception was the nuclear hormone receptor signaling pathways and blood-brain modules we observed to be affected by CIE across all male tissues which was not observed in female mice. There have been very few studies investigating the transcriptional response to alcohol dependence in females [88–90], and while the focus of these studies as well as the present study is not on sex differences, further investigation into the sex-specific responses to CIE is warranted.

There were a number of methodological choices we made for this study, such as the tissue type (whole blood), molecular measurement (mRNA), and model of alcohol dependence (C57BL/6J mice undergoing CIE) that impose limitations and should be considered. There are several accessible tissues that have been compared to brain in previous studies that we could have assessed, including saliva [91] and blood fractions (e.g., exosomes [92, 93], PBMCs [94, 95], or plasma/serum [96–98]). We chose to measure whole blood because it has been shown to contain disease-relevant information for other brain diseases, has less processing steps than for blood fractions, and contains both cellular and extracellular RNAs so would likely capture a signal from each of the aforementioned blood fractions. However, the various blood cell types could have differential responses to alcohol dependence which could be explored using single cell RNA-seq. We chose to measure mRNA, but there are other molecular measurements, including other RNAs such as miRNAs [91, 92, 97, 99, 100], histone modifications (epigenome) [101, 102], microbiome [103, 104], and the metabolome [105–108]. Furthermore, we assayed the transcriptome at a single time point, and gene expression is a highly dynamic

process. Future studies that profile other molecular responses and at different time points will undoubtedly contribute to our understanding of alcohol dependence and incorporating these additional molecular markers into the machine learning models could further improve their predictive ability. In this study we subjected C57BL/6J mice to CIE which is a commonly used animal model of alcohol dependence. There are also genetic rodent models of behavioral characteristics that put one at risk for developing AUD (such as binge-like drinking, or high alcohol preference), that could be analyzed to determine whether blood and brain responses are conserved in those models and if blood could be used to as an objective molecular tool to predict individuals at risk of developing AUD. We employed an animal model in this study to enable the comparison of blood and brain samples from the same subjects (which is not possible in humans). This study established a link between blood and brain responses in an animal model of alcohol dependence. Going forward, we will investigate whether there is a similar signature of alcohol dependence in the blood of human patients with AUD.

Our data indicate that a molecular signature of CIE-induced alcohol dependence can be detected in peripheral blood at least one week after the last alcohol exposure. This signature partially reflects brain pathophysiology and can be used to discriminate CIE and Air mice with a high degree of accuracy. Although it is not possible to draw any clinical conclusions from this dataset, our study establishes a link between blood and brain responses to CIE exposure and demonstrates that blood profiles can distinguish CIE and Air animals, which suggests that blood samples could contain information relevant to alcohol use. These findings represent important first steps in identifying novel biomarkers for AUD and provide critical context for future blood-based biomarker studies. However, more research, including human validation, will be required before genes selected as discriminatory features could become viable biomarkers. We hope this information helps drive objective diagnosis, medication development, and personalized medicine approaches for AUD and other diseases where brain is the primary affected tissue.

## Materials and methods

### Ethics statement

All procedures were IACUC approved and met the guidelines of the National Institute of Health detailed in the Guide for the Care and Use of Laboratory Animals.

### Animals

Adult male and female C57BL/6J mice (Jackson Laboratories, ME) were used in this study (N = 10/sex/group; 40 mice total). One female mouse in the control group and two male mice in the ethanol vapor group died during the experiment leaving N = 19 female mice and N = 18 male mice as the final subjects. Mice were housed at The Scripps Research Institute, four per cage (except during the 2-hr drinking sessions), separated by sex in standard plastic cages under a reversed 12-h light/dark period (lights on at 8:00 PM), with food (Teklad Global 18% Protein Rodent Diet, Envigo) and water available ad libitum.

### Chronic Intermittent Ethanol (CIE) model of alcohol dependence-induced escalation of drinking

CIE was implemented as described in previous studies [25–27, 32, 36, 37, 109–112]. Thirty minutes before the dark cycle, mice were singly housed for two hours with access to two drinking tubes, one containing 15% ethanol and the other containing water. Ethanol and water consumption during these 2-hour periods was recorded. Following this baseline period of two-

bottle choice (2BC) drinking, which lasted 20 days (5 days per week for 4 weeks), mice were divided into two balanced groups with similar distributions of ethanol and water consumption. One group was exposed to intermittent ethanol vapor and the other to control air in identical chambers. The ethanol vapor group was administered 1.75 g/kg ethanol plus 68.1 mg/kg pyrazole to inhibit alcohol dehydrogenase, and then was placed in the chambers to receive intermittent vapor for 4 days (16 hours vapor on, 8 hours vapor off). Once per week, immediately following a 16-hour bout of vapor, mice were removed, and tail blood was sampled for blood alcohol determination. Target blood alcohol levels were 175–250 mg%. Following the fourth day of exposure, mice were allowed 72 hours of undisturbed time. The mice were then given 5 days of 2BC drinking. The control group was injected with 68.1 mg/kg pyrazole in saline and placed in chambers delivering air for the same periods as the ethanol vapor group and then received 2BC testing at the same time as the vapor groups. Mice were subjected to four cycles of vapor or air exposure followed by 5 days of 2BC drinking. All mice then received one final 4-day vapor or air exposure (without 2BC testing). One week later, mice were euthanized and tissue collected for RNA sequencing.

## Blood ethanol analysis for vapor-exposed mice

Approximately 40 μl blood was obtained by cutting 0.5 mm from the tip of each mouse's tail with a clean surgical blade. Blood was collected in capillary tubes and emptied into Eppendorf tubes containing evaporated heparin and kept on ice. Samples were centrifuged and plasma decanted into fresh Eppendorf tubes. Plasma (5 μL) was injected into an Agilent 7820A GC coupled to a 7697A (headspace-flame-ionization). Results were compared with and calibrated using a 6-point serial diluted calibration curve of 300 mg/dl ethanol (Cerilliant E-033).

## Statistical analysis of 2-bottle choice drinking

Average ethanol intake (g/kg) was calculated across 5 drinking days of each week during the baseline-drinking period. During the testing cycles, mice also drank for 5 days; therefore, average drinking across these 5 days was used to represent drinking during each CIE cycle. Differences in drinking were determined by Two Way ANOVA (treatment x time (i.e., cycle)) followed by planned Dunnett's tests to determine whether drinking during each testing cycle was different from baseline drinking levels. We also used a One Way ANOVA (time (i.e., cycle)) for the ethanol vapor group and air group separately followed by planned Dunnett's tests to determine whether drinking during each testing cycle was different from baseline drinking levels. Statistical analysis was implemented with GraphPad Prism 8.3.0 (GraphPad Software, San Diego, CA, USA).

## Tissue collection

Tissue was collected one week following the final vapor/air exposure between 10 AM and 12:30 PM. Mice were anesthetized with isoflurane and approximately 200–250 μl blood was collected from the retroorbital sinus in capillary tubes and emptied into Eppendorf tubes containing 3x volume DNA/RNA Shield (Zymo Research, Irvine, CA, USA). Tubes were immediately vortexed on the highest speed for 30 seconds and remained at room temperature for about one hour before being placed at -80˚C until further processing. Mice were transcardially perfused with ice-cold phosphate buffered saline (40 mL over 15 min). The mice were then decapitated, and the brains removed and flash frozen in liquid nitrogen. Brain and blood samples were kept at -80˚C before being placed on dry ice and shipped to the Dell Medical School (Austin, TX) for further processing.

## Dissection of brain areas

Brain samples were frozen in Optimal Cutting Temperature (OCT) media in isopentane on dry ice and were stored at −80°C until sectioning. On the day of sectioning, brains were transferred to a cryostat set at −6 to −10°C for at least 2 h. Sections (300 μm) were collected from +3.00 to -4.00 mm (AP) relative to bregma and transferred to glass slides that had been pre-cooled on dry ice. The slices between the following coordinates, relative to bregma, were used for prefrontal cortex (PFC): +3.0 mm to +1.70 mm (AP), amygdala (AMY): -1.00 mm to -2.30 mm (AP), hypothalamus (HYP): -1.30 mm to -2.30 mm (AP). We performed bilateral micropunch sampling on a frozen stage (−20 to −25° C) using a 1.00 mm diameter punch (Stoelting Co., item #57397, Wood Dale, IL, USA) to include all sub-regions of the brain areas according to the stereotaxic atlas of Paxinos and Franklin [113]. Punches were stored at −80°C until RNA extraction.

## RNA extraction

Total RNA was extracted from the whole blood samples using the Quick-RNA Whole Blood kit according to manufacturer instructions (Zymo, Irvine, CA, USA). Total RNA was extracted from the brain micropunches using the PureLink RNA Mini Kit (Thermo Fisher Scientific, Waltham, MA, USA) according to manufacturer instructions except that 2-mercaptoethanol was not added to the lysis buffer as this was found to reduce reads in downstream analyses during optimization tests. RNA purity, concentration, and integrity were determined using a NanoDrop 1000 spectrophotometer (Thermo Fisher Scientific, Waltham, MA, USA), Qubit 4 (Thermo Fisher Scientific, Waltham, MA, USA), and Tapestation 4150 (Agilent Technologies, Santa Clara, CA, USA). For quantitative assessment of RNA sequencing, External RNA Controls Consortium (ERCC) RNA Spike-In Mix 1 (Thermo Fisher) were added to samples. We submitted 300 ng total RNA/sample for brain samples and 2000 ng total RNA/sample for whole blood samples to the Genomic Sequencing and Analysis Facility at UT Austin for sequencing. The extracted RNA was of sufficient amount and quality (S9 Table), and subsequent sequencing accurately quantified the RNA as assessed by strong correlations between the actual concentration and observed counts of the ERCC spike-in control ($R^2$ = .90 (PFC), 0.91 (HYP), 0.92 (AMY), 0.85 (blood)).

## RNA sequencing

We surveyed gene expression using TagSeq by constructing libraries directed at the 3' ends of mRNA according to established protocols [114]. About 70% of mRNA from whole blood is globin mRNA, which limits the ability to detect other genes. Therefore, we depleted globin mRNA in the whole blood samples before library construction using the QIAseq Fast Select RNA removal kit (Qiagen, Germantown, MD, USA) according to manufacturer instructions. Before globin depletion, the blood samples were purified using an AMPure XP (Agencourt) bead clean-up as our optimization tests revealed this improved the blood library concentrations. Purified libraries were size-selected to obtain 350–550 bp target cDNA and pooled in equal proportions following relative quantification using a qPCR assay. Pooled sets of libraries were sequenced using an Illumina NovaSeq 6000 generating 1x100 bp single-end reads. Illumina adapters were removed, and lane pools were de-multiplexed according to the barcode assignments by the sequencing facility prior to return of raw data.

## Bioinformatics analysis

The mean read count was approximately 5 million reads per sample. This read depth provides full-coverage detection with TagSeq (which is ~12,000–15,000 unique genes, depending on the tissue). The reads were trimmed, de-duplicated, and quality filtered using custom Perl scripts

from https://github.com/z0on/tag-based_RNAseq. The data were mapped to the GRCm38/mm10 mus musculus reference transcriptome supplemented with the sequences of the ERCC spike-in control mRNAs using bowtie2 [115]. Then samtools [116] and htseq-count [117] were used to sort the mapped reads and convert mapped reads to counts. Read distributions were characterized using the read_distribution.py program from RSeQC package v2.6.2. The genome annotations for the analyses were taken from GENCODE using the main file containing comprehensive gene annotation on the reference chromosomes only. Raw counts were converted to log 2-counts per million (log-CPM) to account for library size differences and then transformed using the voom function from the limma package (version 3.42.2) to account for the heteroscedastic nature of count data [118].

Differentially expressed genes (DEGs) between CIE and Air mice were identified using the limma package (version 3.42.2) [118]. Next the transcriptional responses were parsed into groups of genes whose expression patterns are highly correlated (modules) using the Weighted Gene Coexpression Network Analysis (WGCNA) package (version 1.69) [119]. We built coexpression networks for each tissue individually as we have done previously (e.g., [40, 120, 121]. All detected genes within a tissue were used for network construction. The Pearson correlation coefficient between all pairs of probes across all samples was calculated, and a signed gene coexpression similarity matrix between genes was generated: $S_{ij} = (1 + cor(x_i, x_j)) / 2$. Then, an adjacency matrix, $a_{ij} = S_{ij}^\beta$, was used to assess gene connections. Power ($\beta$) was chosen so that the resulting network exhibited approximate scale-free topology (for male PFC, AMY, HYP, and blood $\beta$ = 6, 8, 7, and 10, and for female PFC, AMY, HYP, and blood $\beta$ = 5, 9, 11, and 6). Next, the topological overlap measure (TOM) was used to calculate the relative interconnectedness of a gene pair. Average linkage hierarchical clustering was applied to produce a dendrogram based on the topological overlap dissimilarity ($1 - TOM$). Branches of the tree were cut using a dynamic tree cut algorithm to detect modules (deepSplit = T, minimum module size = 100, cut height = 0.99). Similar modules were merged together (cutHeight = 0.25). The terms "modules," "clusters," and "gene networks" are used interchangeably in the manuscript and refer to groups of genes with highly correlated expression levels across samples.

Coexpression modules were said to be related to alcohol dependence if: (1) the module contained more DEGs than expected by chance (hypergeometric test $p < 0.05$); or (2) the module eigengene was correlated with alcohol dependence status, alcohol consumption, or alcohol preference during the voluntary drinking sessions. The module eigengene is defined as the first principal component of the expression matrix of the corresponding module and can be thought of as the summary of gene expression within each module.

Blood and brain are heterogenous tissues comprised of multiple cell types. To gain insight into functional roles for particular cell types, we determined whether cell type-specific genes were enriched in the DEGs and alcohol-related modules using cell type signatures from the literature and the userListEnrichment function from the WGCNA package in R. The brain cell type markers included six major cell types in the brain: astrocytes, endothelial cells, microglia, neurons, oligodendrocytes, and oligodendrocyte progenitor cells (OPCs) [122]. The immune cell type markers included seven immune populations: B cells, plasma cells, monocytes, macrophages, neutrophils, NK cells, and T cells [123]. Terms with Bonferroni-corrected hypergeometric $p < 0.05$ were considered significantly enriched within the dataset.

To determine what functional impact the observed transcriptional responses might have, we performed enrichment analysis on the DEGs and alcohol-related modules using the Ingenuity Pathways Analysis knowledgebase (IPA, Ingenuity Systems, www.ingenuity.com), a web-based software application. We performed a core analysis for the DEGs and modules of interest in IPA using default settings, except that expressed transcripts were used as the

background population for the right-tailed Fisher exact test (FET) calculations. Terms with FET $p < 0.05$ were considered significantly enriched within the dataset.

## Blood and brain gene expression comparisons

A major goal of this study was to compare the transcriptional response to CIE-induced alcohol dependence between blood and brain. We compared DEGs in whole blood and each brain area and identified overlapping DEGs between tissues. We determined whether a greater number of DEGs were shared between blood and brain than expected by chance using the hypergeometric test (hypergeometric $p < 0.05$ was considered significant). In addition to comparing the DEGs, we also compared the gene coexpression modules that were related to CIE between blood and brain following previously described approaches [78, 120, 121, 124, 125]. Briefly, for each pair of networks, the overlap between all possible pairs of modules was calculated, and the significance of module overlap was assessed using a one-sided hypergeometric test. Blood and brain modules sharing a significant number of genes (hypergeometric $p < 0.05$), were considered conserved and we refer to them as blood-brain modules throughout the manuscript. The software Cytoscape (http://www.cytoscape.org/) was used to visualize the comparisons and create a meta-network of highly overlapping CIE modules.

Another goal of the study was to determine whether blood gene expression levels are correlated with (i.e., predictive of) brain gene expression levels. Most previous studies comparing blood and brain gene expression in humans have used blood and brain data from different individuals (brain samples collected postmortem and blood samples collected from living patients). To make our results more comparable to previous studies, we analyzed our data in a similar manner (between-subjects design). For the between-subjects correlation, each point in the scatterplot represented a gene, and we plotted the normalized expression level of the gene averaged across subjects. We then calculated the Spearman correlation coefficient between blood and brain normalized gene expression levels using the corr.test function from the psych package (version 2.0.7) in R Studio. We compared the correlations between brain and blood for males and females using the cocor package (version 1.1–3) in R Studio [126].

We measured gene expression levels for brain and blood for the same animal which permits a within-subjects comparison. For the within-subjects analysis, we plotted the normalized expression level for a gene in brain and blood for the same subject, where each point in the scatterplot represented a subject. We then calculated the Spearman correlation coefficient between blood and brain normalized gene expression levels for each gene using the corr.test function from the psych package (version 2.0.7) in R Studio. There were over 10,000 correlation calculations performed, one for each gene. To account for multiple comparisons, we used Holm-Bonferroni correction, and corrected $p < 0.05$ was considered significant [127].

## Classification algorithms and parameter selection

We determined how well blood gene expression levels could discriminate between CIE and Air subjects using three classification models that are exemplary at identifying patterns or trends in 'omic' data and include measures of variable importance to enable interpretation of the models: logistic regression (LR) with elastic net regularization, random forest (RF), and partial least squares discriminant analysis (PLSDA). Repeated cross validation was used to choose the optimal hyperparameters (5-fold cross validation repeated 10 times) for each model. The optimal value for the RF parameter mtry (the number of variables available for splitting at each tree node) was chosen to be 638 for males and 197 for females. The LR regularization parameters are alpha (this parameter balances the amount of emphasis given to minimizing Residual Sum of Squares versus minimizing sum of square of coefficients) and lambda

(regularization penalty). Alpha was 0.1 and lambda was 0.135 for males. Alpha was 1 and lambda was 0.341 for females. The PLS parameter ncomp (number of components to include in the model) was three for males and four for females. Model training and evaluation was implemented with the MLSeq package (version 2.8.0) in R version 4.0.3 [128].

## Classifier performance evaluation

To evaluate the performance of the classifiers in assigning the correct label to each subject (CIE versus Air) we used 5-fold cross validation. MLSeq takes a matrix of raw counts as the input and performs normalization within-fold so that the normalization of the test fold is performed using coefficients estimated from the training folds. The process of randomly splitting samples into training/test folds, training the models, and then testing performance was repeated 10 times to obtain estimates of model performance. The performance metrics were averaged across the repeated folds. The performance metrics we calculated were accuracy (percentage of correct assignments), area under the Receiver Operating Characteristic curve (AUROC), and Matthews correlation coefficient (MCC). Given a test set and a specific classifier, each decision can be categorized as one of the following: (1) a positive example classified as positive (true positive; TP), (2) a positive example misclassified as negative (false negative; FN), (3) a negative example classified as negative (true negative; TN), (4) a negative example misclassified as positive (false positive; FP). After creating a Contingency Table, a 2×2 matrix with the columns as true classes and the rows as the hypothesized classes, we calculated accuracy (the fraction of predictions the model assigned correctly) as follows: $\frac{TP+TN}{TP+TN+FP+FN}$, sensitivity (the True Positive Rate, or the proportion of positives that are correctly identified): $\frac{TP}{TP+FN}$, specificity (the True Negative Rate, or the proportion of negatives that are correctly identified): $\frac{TN}{TN+FP}$, and the False Positive Rate: $1 - \frac{TN}{TN+FP}$. To plot the ROC curve, we calculated the False Positive Rate and True Positive Rate under different classification thresholds, then quantified the performance of the classifiers by calculating the area under the ROC curves (AUROC) using the roc function from the pROC package (version 1.16.2) in R. The higher the AUROC, the better the classifier performance. AUROC of 0.5 is random, 1.0 is perfect, and 0.7–0.8 is generally considered high performance [129].

We also calculated the Matthews correlation coefficient (MCC) as: $\frac{TP \times TN - FP \times FN}{\sqrt{(TP+FP)(TP+FN)(TN+FP)(TN+FN)}}$. MCC is essentially a correlation coefficient between the observed and predicted binary classifications; it returns a value between −1 and +1. A coefficient of +1 represents a perfect prediction, 0 no better than random prediction and −1 indicates total disagreement between prediction and observation.

## Variable importance measures

Logistic regression, random forest, and partial least squares discriminant analysis provide measures of feature importance, which enable the identification of which genes were the most useful for discriminating between the CIE and Air subjects. When building a random forest, useful genes will split the mixed labeled nodes into pure single class nodes. This measure is termed Gini importance. In PLSDA, the loading vectors (which are coefficients assigned to each gene to define each component) are obtained so that the covariance between a linear combination of the genes and the class label is maximized. The variable importance measure for PLSDA is based on the weighted sums of the absolute regression coefficients (the loading vectors). The weights are proportional to the reduction in the sums of squares. For LR, the importance measure is the regression coefficient. Importance measures were extracted from the final models using the varImp function from the caret package (version 6.0–86).

## Supporting information

**S1 Fig. Comparison of blood and brain mean gene expression levels.** The scatterplots display the relationship between blood (x-axis) and brain (y-axis) mean gene expression levels for male (left) and female (right) mice. Each point in the scatterplot represented a gene, and the normalized expression level of the gene averaged across subjects (irrespective of treatment) are plotted. We then calculated the Spearman correlation coefficient between blood and brain normalized gene expression levels, and this value is displayed in the plots (rho).
(TIF)

**S2 Fig. Correlation of blood and brain gene expression levels with alcohol consumption.** The scatterplots display the relationship between alcohol consumption in the final limited access two bottle choice drinking test (y-axis) and blood or brain gene expression levels (x-axis) for the genes presented in Fig 4 for female (A) and male (B) mice. Each point in the scatterplot represented a subject. We then calculated the Pearson correlation coefficient between the normalized gene expression levels and alcohol intake and this value and associated p-value is displayed under the x-axis.
(TIF)

**S1 Table. The full differential expression results.**
(XLSX)

**S2 Table. The cell type enrichment analysis results of the DEGs.**
(XLSX)

**S3 Table. The full IPA pathway enrichment analysis and upstream regulator analysis for the differentially expressed genes (CIE versus Air, p $<$ 0.05).**
(XLSX)

**S4 Table. The full IPA upstream regulator results for the genes in the blood-brain modules.**
(XLSX)

**S5 Table. The full IPA pathway enrichment results for the genes in the blood-brain modules.**
(XLSX)

**S6 Table. Comparison of the correlation coefficients between blood and brain mean gene expression levels in females and males.**
(XLSX)

**S7 Table. List of genes whose expression levels are correlated between brain and blood (Holm-corrected p$<$0.05).**
(XLSX)

**S8 Table. The full IPA upstream regulator results for the genes whose expression levels are correlated between brain and blood.**
(XLSX)

**S9 Table. RNA quality and quantity assessments.**
(XLSX)

## Acknowledgments

We thank Dr. Dennis Wylie for providing feedback on the machine learning component of the analysis. We would also like to acknowledge The Scripps Research Institute's Animal Models Core for providing the research space, tools, and equipment needed for running the chronic intermittent ethanol exposure procedure.

## Author Contributions

**Conceptualization:** Laura B. Ferguson.

**Formal analysis:** Laura B. Ferguson.

**Funding acquisition:** Laura B. Ferguson, R. Dayne Mayfield, Robert O. Messing.

**Investigation:** Laura B. Ferguson, Amanda J. Roberts.

**Methodology:** Laura B. Ferguson, Amanda J. Roberts.

**Project administration:** Laura B. Ferguson, Amanda J. Roberts, R. Dayne Mayfield, Robert O. Messing.

**Supervision:** R. Dayne Mayfield, Robert O. Messing.

**Visualization:** Laura B. Ferguson.

**Writing – original draft:** Laura B. Ferguson.

**Writing – review & editing:** Laura B. Ferguson, Amanda J. Roberts, R. Dayne Mayfield, Robert O. Messing.

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
