## [Decision Letter · Decision Letter 0]

7 Oct 2021

Dear Dr. Ferguson,

Thank you very much for submitting your manuscript "Blood and brain gene expression signatures of chronic intermittent ethanol consumption in mice" for consideration at PLOS Computational Biology.

As with all papers reviewed by the journal, your manuscript was reviewed by members of the editorial board and by several independent reviewers. In light of the reviews (below this email), we would like to invite the resubmission of a significantly-revised version that takes into account the reviewers' comments.

We cannot make any decision about publication until we have seen the revised manuscript and your response to the reviewers' comments. Your revised manuscript is also likely to be sent to reviewers for further evaluation.

Sincerely,

Atle van Beelen Granlund, Ph.D

Guest Editor

PLOS Computational Biology

Ilya Ioshikhes

Deputy Editor

PLOS Computational Biology

Reviewer's Responses to Questions

**Comments to the Authors:**

Reviewer #1: This manuscript by Ferguson and colleagues performs a rigorous comparison of total and ethanol-responsive gene expression profiles in blood versus several brain regions in C57BL/6J mice that have been treated with a chronic intermittent ethanol vapor + consumption model of ethanol exposure. While a number of other studies have done somewhat similar work for other disease states or with postmortem tissue, this is apparently the first well-controlled animal model study to conduct such whole genome profiling of brain vs. blood with ethanol treatment. The experimental design and methodology are well described and the results are presented in a thorough and clear fashion. The study presents a comprehensive analysis at the level of individual genes, expression networks, biologic functional groups, and a “network of networks” approach to assess overlap of brain vs. blood expression networks. Finally, the authors also present the use of 3 different predictive models to assess the ability of blood expression patterns to predict the treatment status of the mice (alcohol exposed vs. control). Overall, this is a very thorough and well-conducted study of importance to the field in terms of identifying approaches for blood biomarker utilization in assessment of alcohol use disorder clinical status. The approaches utilized should lead to future important work with potential clinical implications.

Weaknesses of the work were few and included:

1) The classification analysis appeared impressive but, in retrospect, seems somewhat superficial with limited direct clinical application as currently developed, although future iterations may be very fruitful. Predicting whether a mouse has been exposed to weeks of high dose ethanol vapor, based upon blood expression patterns, is a rather low bar and the clinical equivalent seems even less significant. It is usually very clinically evident, by history, physical exam or routine laboratory studies, that an individual is a severe case of alcohol use disorder. Whether, the blood expression profiles might detect milder forms of alcohol use disorder and thus be more clinically useful would require either an additional control group (e.g. no ethanol intake at all) or a larger number of animals to allow analysis of low vs. high drinkers. The investigators should at least include some of these issues in the Discussion.

2) It is surprising that despite the extremely detailed analytical scheme and bioinformatics analysis, that the investigators did not include a co-analysis of their brain and blood network patterns vs. networks/genes that have been published for autopsied human tissue or GWAS studies on AUD or alcohol consumption. Such cross-species studies would add clinical relevance.

3) Some of the individual gene correlations in blood vs. brain were quite impressive (Fig. 4) and a number of these were regulated by ethanol vapor exposure. The multiple heat shock proteins are particularly striking as they have been identified as ethanol-responsive genes dating back to the early 1990’s. It would be an additional piece of useful information to see a plot of the expression for some of these genes (in brain or blood) vs. the ethanol consumption levels for ethanol control or vapor-exposed animals – to identify the impact of oral consumption vs. just the vapor.

4) Although perhaps a minor semantic issue, the extensive use of the term “alcohol dependent” in describing the animals exposed to ethanol vapor (CIE) is somewhat unsettling. The terminology “alcohol dependence” usually refers to a clinical state consisting not just of the presence of withdrawal symptoms upon ethanol abstinence, but of obsessive ethanol seeking behavior, amongst other clinical signs. The CIE mice do show some increased ethanol consumption and other investigators have shown occurrence of increased withdrawal seizures and evidence of tolerance – but this reviewer is not certain that the term “alcohol dependent” is entirely appropriate.

Reviewer #2: The submission entitled “Blood and brain gene expression signatures of chronic intermittent ethanol consumption” describes gene expression changes in alcohol-dependent and control mice in three brain regions, namely: prefrontal cortex, hypothalamus and amygdala as well as in blood; and analyses how such changes correlate between brain and blood using a within-subject approach. Their results start to address a gap in our knowledge about how blood gene expression profiles can be used to diagnose AUDs, and have the potential to aid in the development of personalized and more effective therapeutic interventions. The manuscript is well written, the limitations clearly stated, and the data was thoroughly analyzed; RNA sequencing information was made publicly available in Gene Expression Omnibus.

The following revisions are recommended:

- Figure 2A is missing labels over: male AMY increase, female PFC decrease, male PFC increase, male HYP increase, female blood increase, male blood increase.

- DEGs and PBMCs need to be defined the first time they appear.

- In the introduction, explain the specific reasons why amygdala, hypothalamus and prefrontal cortex where chosen for this study. In the methods, please clarify what specific sub-regions where collected for each brain region, was the 1mm punch used to collect from all sub-regions or was there a focus on specific ones?.

- In the results section, Page 10, lines 14-18 (Figure 2A), please clarify neuronal changes in the female hypothalamus; what are the differences between the neuronal genes going up and those going down?.

- In the introduction and methods, please clarify if the animals where drinking after the 4th CIE cycle and explain the reason tissue was collected 1 week after CIE, presumably to avoid having alcohol on-board at collection time?. Typically mice will still show increased ethanol consumption on the last test-drinking day, but if they have not been given access to alcohol for a whole week their withdrawal/motivation state can’t be extrapolated to (or assumed to be) that of a test-drinking week.

Discussion

- Although the main focus of this study is to identify gene expression signatures common between blood and brain, it would be important for the authors to discuss how the brain DEGs found in dependent mice compare to those in human alcohol-dependent individuals in those same regions, according to the literature.

- The discussion would be strengthened if the expectations for developing better, more personalized, pharmacological interventions using blood gene sequencing are described and how the current results contribute to that end. How could this technology be used to identify individuals with high risk of AUDs, and help develop prophylactic interventions for the development of dependence?. Future studies, could analyze blood RNA seq before and after CIE treatment and correlate gene expression changes with the amount of drinking escalation within subjects.

Reviewer #3: In this manuscript, the authors profiled and analyzed transcriptomes of three brain regions and blood in C57BL/6J mice one week after chronic intermittent ethanol (CIE) exposure. They found the transcript level between blood and three brain regions is highly preserved. Despite the small overlap between blood and brain DEGs, considerable overlap at gene network level was observed. The authors then used three machine-learning algorithms to predict alcohol dependence status with blood transcriptomic data. While these results are intriguing, the authors should submit their work to alcohol or addiction relevant journals instead of PLoS Computational Biology as all analyses in this study are conventional and do not involve any methods/algorithms/tools development.

Major comments:

1. One of the major goals of this study was to compare the transcriptional response to alcohol dependence between the blood and different brain regions. However, the authors only provide Venn diagrams (Figure 2B) to show the overlap between differentially expressed genes (DEGs) in blood and brain regions without taking into consideration the magnitude of change in gene expression (e.g., fold change). This can be misleading because the analysis used a loose cutoff (see comment #2).

2. For the DE analysis, no multiple testing control and a very loose cutoff level (nominal p < 0.05) were used. The reviewer suspects that there is a very low overlap between DE genes in blood and brain regions if adjusted p.value < 0.01 is applied.

3. It is not clear how the authors performed co-expression module preservation between blood and brain regions. Why not using the module preservation Z-summary statistics provided by WGCNA?

4. For the three machine-learning algorithms, it is not clear how many features were selected and what feature selection methods were used?

**Have the authors made all data and (if applicable) computational code underlying the findings in their manuscript fully available?**

Reviewer #1: Yes

Reviewer #2: Yes

Reviewer #3: **No: **Not applicable

PLOS authors have the option to publish the peer review history of their article (what does this mean?). If published, this will include your full peer review and any attached files.

Reviewer #1: No

Reviewer #2: No

Reviewer #3: **Yes: **Xusheng Wang
---

## [Decision Letter · Decision Letter 1]

3 Jan 2022

Dear Dr. Ferguson,

We are pleased to inform you that your manuscript 'Blood and brain gene expression signatures of chronic intermittent ethanol consumption in mice' has been provisionally accepted for publication in PLOS Computational Biology.

Best regards,

Atle van Beelen Granlund, Ph.D

Guest Editor

PLOS Computational Biology

Ilya Ioshikhes

Deputy Editor

PLOS Computational Biology

Reviewers all agree that you have addressed all their comments satisfactorily, and that your manuscript thus will be accepted for publication. Below you will find the final comments by the Reviewers to you revised manuscript.

Reviewer's Responses to Questions

**Comments to the Authors:**

Reviewer #1: The authors have responded thoroughly to all concerns previously raised by this reviewer. The authors are to be commended for the thoroughness of their analyses, the full documentation of their methods (including suppl. material) and the posting of materials on open access sites such as GEO and GitHub. Also the Shiny app for exploration of their data is also an important contribution.

Reviewer #2: Thank you for the opportunity of reviewing this revised article submission. The authors have addressed all my previous comments throughly and satisfactorily, by providing additional data, clarifying methods and expanding the discussion. My only remaining, minor, question is whether the data from the 3 drinking tests post CIE show higher increase in EtOH consumption on the last test day, as it was seen in reference #27. The reason for this question is that this might not always be the case; others (ie PMID: 26935824) have not shown this trend and, although my lab has previously only reported weekly average consumption, the first day of every test week shows the highest consumption (unpublished observation). I believe this manuscript provides important insights about chronic alcohol use-induced signature changes in gene expression, and their blood brain correlations, and it is now aceptable for publication.

Reviewer #3: The reviewer's comments have been fully addressed.

**Have the authors made all data and (if applicable) computational code underlying the findings in their manuscript fully available?**

Reviewer #1: Yes

Reviewer #2: Yes

Reviewer #3: Yes

PLOS authors have the option to publish the peer review history of their article (what does this mean?). If published, this will include your full peer review and any attached files.

Reviewer #1: No

Reviewer #2: No

Reviewer #3: **Yes: **Xusheng Wang

---

## [Editor Report · Acceptance letter]

24 Jan 2022

PCOMPBIOL-D-21-01250R1 

Blood and brain gene expression signatures of chronic intermittent ethanol consumption in mice

Dear Dr Ferguson,

I am pleased to inform you that your manuscript has been formally accepted for publication in PLOS Computational Biology. Your manuscript is now with our production department and you will be notified of the publication date in due course.

With kind regards,

Olena Szabo
